# The Impact of Force Factors on the Benefits of Digital Transformation in Romania

Sorinel Căpușneanu [1,*], Dorel Mateș [2], Mirela Cătălina Tűrkeș [3], Cristian-Marian Barbu [4], Adela-Ioana Staraș [5], Dan Ioan Topor [6], Laurențiu Stoenică [4] and Melinda Timea Fűlöp [7]

1   Faculty of Finance-Banking, Accountancy and Business Administration, Titu Maiorescu University, 040051 Bucharest, Romania
2   Faculty of Economic Sciences, West University of Timisoara, 300223 Timișoara, Romania; dorel.mates@e-uvt.ro
3   Faculty of Economics and Business Administration, Dimitrie Cantemir Christian University, 040042 Bucharest, Romania; mirela.turkes@ucdc.ro
4   Faculty of Management-Marketing, Artifex University, 060754 Bucharest, Romania; cbarbu@artifex.org.ro (C.-M.B.); lstoenica@artifex.org.ro (L.S.)
5   National Institute for Chemical Pharmaceutical Research and Development (INCD-ICCF), 031299 Bucharest, Romania; adela_staras@yahoo.com
6   Faculty of Economic Sciences, 1 Decembrie 1918 University, 510009 Alba-Iulia, Romania; dan.topor@uab.ro
7   Faculty of Economic Sciences and Business Administration, Babes-Bolyai University, 400084 Cluj-Napoca, Romania; melinda.fulop@econ.ubbcluj.ro
*   Correspondence: sorinelcapusneanu@prof.utm.ro; Tel.: + 40-72-039-8735

**Abstract:** The digital transformation has produced changes in all existing areas of activity worldwide. There are many factors that can influence the intention to use Industry 4.0 processes and solutions and change the behavior of organizations and their business models. The aim of this study is to validate the econometric model on assessing the significant impact of distinct factors on the intention to use Industry 4.0 processes and solutions, the benefits of digital transformation perceived by organizational management and the differences between distinct groups analyzed. The research method used within the quantitative study was the sample survey, using the online questionnaire as a data collection tool. Three hundred forty-seven valid questionnaires were collected and the response rate of the respondents was 64.25%. A new structural model was generated based on the elements of the Unified Theory of Acceptance and Use of Technology (UTAUT). The results of the study indicated that Perceived competitiveness and Perceived risk have a significant impact on Intention to Use Industry 4.0 processes while Perceived vertical networking solutions and Perceived integrated engineering solutions have a significant influence on the Intention to Use Industry 4.0 solutions. In conclusion, there is a positive and significant association between Intention to Use Industry 4.0 solutions and Benefits of Digital Transformation.

**Keywords:** digital transformation; Industry 4.0; UTAUT; value creation; business environment

## 1. Introduction

Today, digital transformation is becoming the most important and necessary component of human life in almost every business that seeks growth, expansion, quality and sustainability [1]. Due to the phenomenon of globalization and pressures from customers and the wave of digitalization, companies face fierce competition trying to survive and gain various competitive advantages [2,3]. Faced with the maturation of digital technologies and the penetration of digitalization in all markets [4], in order to govern these complex transformations, companies must establish effective management practices without affecting the organizational structure or nomenclature of processes and products [5]. Although there are many possibilities to implement new digital technologies, not all companies have managed to keep up with technological progress [6], encountering many difficulties in

changing or adapting business models [7], many of them feeling threatened by the new digital wave [8]. To cope with the digital transformation, organizations need to develop adaptive capabilities based on the current business context and customer needs. In order to maintain their competitiveness, organizations need to think about reinventing business models, how future businesses will operate as digital transformation becomes essential and vital in the future [9].

Through our study, we try to find some answers to the questions: What are the forces that influence the intention to use processes and solutions in Industry 4.0? How can they radically change the business models of organizations, generating new challenges and benefits? The aim of the study is to validate the econometric model assessing the significant impact of the distinct factors on the intention to use Industry 4.0 processes and solutions, the benefits of the digital transformation perceived by the organizational management in Romania and the existing differences between the distinct groups analyzed. The uniqueness of this Romanian study is highlighted by the five central objectives of the research, namely: (a) identifying the forces that influence the organization's management intention to use Industry 4.0 processes; (b) identification of factors that have a direct impact on the intent to use Industry 4.0 solutions; (c) identifying the main benefits of digital transformation; (d) measuring the robustness of the structural model and (e) highlighting the differences between the groups analyzed using the proposed structural model.

The stage of implementation of Industry 4.0 in Romania is incipient [10]. Currently, Romania has financing lines dedicated to the implementation of new innovative concepts and technologies such as Industry 4.0, 3D Printing and Open Innovation, which are supervised by the Ministry of Economy. Some companies operating in Romania, such as Deloitte, DHL, DB Schenker, Microsoft and Oracle, have made investments in warehouse management systems (Warehouse Management System) or transportation management systems (Transportation Management System), but also applications that help track orders and transparency in supply chains. The largest companies have developed digital platforms [11], adopted IoT solutions [12] and augmented reality, developed 4G and 5G networks, thus producing a revolution in IoT. There are significant development opportunities for Romania in terms of Industry 4.0 and the direction indicated becomes clear, which is why companies need to understand the importance and urgency of digitalization on which the success or failure of many of them depends. We believe that this study will facilitate the transition of organizations to the implementation of Industry 4.0 and the benefits will provide managers with new directions for identifying priority criteria and real opportunities to make appropriate decisions in the business environment.

The content of the article is structured in accordance with the proposed purpose and objectives, with a brief presentation of the literature in Section 2, a description of the research methodology and data source in Section 3, and presentation of the empirical results in Section 4. Finally, discussion of the results obtained, conclusions and limitations of the study are presented in Section 5.

## 2. Literature Review

### 2.1. Digital Transformation Concepts and Drivers

Since the advent of the term digital transformation in the 1970s [13], the concept has undergone a number of interpretations, according to scholars and specialists, being interpreted through the prism of stakeholders [14] as follows: (a) an organizational strategy formulated and executed by the capitalization of digital resources to create differential value [15]; (b) the extent to which an organization engages in any IT activity [16] and is characterized by the use of new digital technologies to enable significant business improvements [17]; (c) the use of new digital technologies, such as social media, mobile, analytics or embedded devices, to improve business such as customer experience, streamline operations or create new business models [18] for the purpose of encouraging the performance and coverage of a company [2]; (d) digital transformation is the deliberate and continuous digitalization of a company's evolution, business model, process of ideas

or strategic and tactical methodology [19]; (e) a part of the organization's strategy that includes networking between actors such as businesses and customers along value-added chain segments [20,21]; (f) a profound and accelerated transformation in terms of processes, activities, skills and models, which benefit from changes and opportunities [22] being based on technologies such as cloud, mobile, social and big-data analysis. To ensure the successful implementation of the digital transformation, organizations and companies need to ensure that they are fully aware of the failures or risks they face over a period of time. For this reason, organizations need to identify those key factors that influence digital transformation in an organization [23]: digital strategy, digital maturity, business model and digital technology used.

An effective digital strategy is to reconfigure a company's business by taking advantage of the information that technology brings and enables, and not to acquire and implement appropriate technology [24]. Unlocking the potential of digital transformation is achieved by capitalizing on strategy, culture and leadership, and on a balanced approach to technology, data integration and organizational change [25]. To achieve significant benefits, organizations need to integrate with digital technology and other elements such as stakeholders, processes and related functions in order to create a profitable and advantageous business environment in the long run. In this regard, two digital strategies have been identified that could be used by organizations or companies: (1) strategy focused on customer involvement and (2) strategy focused on digital solutions [26]. The customer engagement strategy is based on establishing customer loyalty and trust by providing them with innovative, personalized and integrated experiences, such as omnichannel networks that facilitate the exchange of direct information between organizations and customers, anticipating their needs, using a volume of big-data analysis. The digital solutions strategy involves combining existing skills with the capabilities offered by digital technologies, trying to collect additional information and creating added value by not focusing on customer requirements.

Digital maturity can be explained by digital technology which is the tool for the successful transformation of processes, talents and business models [27], and which generates added value to an organization. To ensure the success of digital maturity, the organization must also add to the digital transformation and digital capabilities, the strategies, organizational culture and the human factor. The business model has received various interpretations from specialists: (a) which business are operated [28]; (b) creating value that allows a differentiation from competitors, strengthening customer relationships and achieving a competitive advantage [29]; (c) a model framework to characterize digital multisided platforms [30]. Digital transformation is based on the business model of the organization or company by adding the value of the chains and network of different digital actors with which it is in business relationships, such as IS infrastructure or digital resources.

The Industry 4.0 concept is a branch of material production in which innovative elements and technologies are integrated (Big Data and Analytics), various devices are implemented (cyber-physical systems (CPS), Internet of Things, cloud computing) and functional aspects are addressed as services, ensuring their constant communication and relationship [31]. The main applications of Industry 4.0 are: Internet of Objects (IoT), Embedded Software, Big Data and Data Analytics, Machine-to-Machine Communication (M2M), Cloud Solutions, Intelligent Robot Automation Systems, End-to-End Software Integration, Augmented Reality, Simulation, Additive Production (3D Printing), Cyber Security, Central Monitoring and Control (SCADA), Mobile Devices, Smart Sensors (RFID, QRC, BARCODE), Smart Objects and Remote Control [32].

Information Communication Technology (ICT) could improve and have a strong impact on the processes related to the delivery of products and services [33]. ICT acts as a facilitator of digital transformation by strengthening the digital strategy when using customer involvement and the digitized solution. In addition to ICT, there are other major technological facilitators such as digital data, automation, networking and digital customer access [21]. These stimulate the value of digitalization, the availability of data,

the automation of production processes, the interconnection of the value of chains and the creation of digital interfaces for customers, transforming the business model and reorganizing the entire industry [34].

### 2.2. UTAUT and Development of Variables

For the implementation of any type of technology, the study of human behavior is used as part of the information system and the most significant is the unified theory of acceptance and use of technology (UTAUT) [35] that helps predict technology acceptance in organizational implementations. Developed by reviewing and consolidating several models (motivated action theory, TAM 1, model of PC use (MPCU), planned behavior theory, motivational model, TAM and TPB combined, social cognitive theory and innovation diffusion theory), UTAUT is based on four key exogenous or independent variables that affect the behavioral intent to use the technology, namely: (a) performance expectancy, (b) effort expectation, (c) behavioral social influence and (d) facilitation conditions with a direct impact on use of technology [36]. Compared to the existing studies of the specialists [37,38], we considered the following variables as representative for our Romanian study: Intention to Use Industry 4.0 processes (IUP) with the three influences (Perceived competitiveness (PC), Perceived opportunities (PO), Perceived risk (PR)), Intention to Use Industry 4.0 solutions (IUS) with three influences (Perceived vertical networking solutions (VS), Perceived horizontal integration solution (HS), Perceived integrated engineering solution (IS)) and Benefits of Digital Transformation (BDT). By creating a new conceptual model based on these variables, our study covers some existing gaps in the literature and opens new opportunities for future research in both academia and business. The relationship between the four exogenous variables of the UTAUT model and the three variables that inform the proposed model (IUP, IUS and BDT) are described below.

### 2.3. Intention to Use Industry 4.0 Processes (IUP)

The intention to use Industry 4.0 processes is related to the three factors: perception of competitiveness, perception of opportunities and perception of risks.

#### 2.3.1. Perceived Competitiveness (PC)

Competitiveness is the distinguishing factor between companies that want to be best highlighted in a particular market segment or globally, which is why their competence in technology, costs and changes must be reassessed. Companies need to develop innovations based on these skills, or they can create differences compared to their competitors [39]. Through competitiveness, companies must offer a variety of high-quality products and services and ensure that these aspects have been perceived by customers. IT is an integral part of every product [40] and this produces structural changes affecting every phase of life in the product manufacturing cycle creating business opportunities [41]. These smart products will create product differentiation, customer segmentation, dynamic pricing, value-added services and closer customer relationships [42]. With the predominant world of digitalization and the continuous transformation of the business environment, competition will continue to grow. From the point of view of competitiveness, the challenges brought by the digital transformation are focused on costs, technological knowledge and IT systems [43], and will take into account the vertical or horizontal integration.

#### 2.3.2. Perceived Opportunities (PO)

Opportunities for using Industry 4.0 include organizational efficiency, organizational agility, manufacturing innovation, product quality and safety and process improvement. Having the right technology tools that work together can streamline workflow and improve productivity. By automating many manual tasks and integrating data across the organization, it allows team members to work more efficiently. The survival of an organization on the market is given by obtaining efficiency and effectiveness in organizational performance [44]. Organizational efficiency involves the development of dynamic capacities

for adaptation, integration and reconfiguration to achieve internal and external competitiveness [45]. The agility of an organization translates into the ability of an organization to develop with a quick response to all situations of competitive uncertainty in order to transform this opportunity into innovative quality products and services [44,46–48]. The agility of the organization will result in a long-term increase in its performance.

Throughout the life of the products, stakeholders (product design companies, manufacturers, testing laboratories, retailers) assume the inevitable responsibility for safety [49] and product quality testing in various manufacturing phases [50], being the most important factor in the manufacturing and retail trade [51]. Manufacturing companies will produce exactly what customers need [52]. Optimizing supply chain processes by using real-time data will result in reduced inventory for all items in the supply chain [53]. Awareness of smart products will result in fewer quality defects, low waste rate, reliable production systems and the overall level of quality of the manufacturing process will increase [54]. Having the product life data available creates the opportunity to continuously improve the product quality [55] and create pleasant customer experiences, knowing that they do not mind paying more for it [56].

### 2.3.3. Perceived Risk (PR)

In a competitive environment that results in a transparent business ecosystem for an online platform [57], the implementation of Industry 4.0 facilitates data exchange when using vertical, horizontal and end-to-end integration [58]. This transparency can expose organizations or companies to cyber-security issues, such as data manipulation processes, cyber attacks, know-how protection, data protection, product protection and other sensitive data [59]. Only by implementing an appropriate strategy and adopting tough cyber-security measures will the organization be able to meet the challenges of cyber attacks and maintain its true image in front of customers, competition and all stakeholders. At the same time, organizations or companies that set the standards of a platform will prevent other competing organizations from accessing information, taking them out of business [60] and creating business models and designing new business strategies.

### 2.4. Intention to Use Industry 4.0 Solutions (IUS)

The intention to use Industry 4.0 solutions is perceived in terms of three factors: vertical networking solutions, horizontal integration solutions and integrated engineering solutions.

### 2.4.1. Perceived Vertical Networking Solutions (VS)

By using hierarchical subsystems necessary to create reconfigurable and flexible manufacturing systems through vertical integration, an organization will develop integrated information subsystems that will help reduce resource losses and improve organizational efficiency [58,61]. The integration of the aggregation and hierarchy of value creation levels within an organization is achieved through vertical integration. Satisfying customer needs through quick responses by offering innovative and customized products is achieved by intelligently cross-linking the various subsystems within the organization (e.g., production and sales). The digital transformation will reduce greenhouse gas emissions; this is due to traceable carbon footprint data or emission data that will be strictly monitored by an algorithm that controls different parameters in the case of vertical integration [62].

### 2.4.2. Perceived Horizontal Integration Solution (HS)

The organizational result can also be improved by efficiently manipulating supply chains as a result of the horizontal integration of the existing information subsystem along the value chain of products and services between organizations [63]. A cross-linking is also ensured between companies and enterprises within the value supply chain to meet customer needs when using horizontal integration. Responsiveness and deep availability of information will increase while safety and quality issues will decrease, which can be

achieved by using product safety data and improving system quality management, self-regulation and automatic monitoring of quality characteristics, as a result of implementing vertical, horizontal or end-to-end integration [64]. By optimizing processes, vertical and horizontal integration will result in a shorter time to market and a considerable reduction in delivery times [65].

### 2.4.3. Perceived Integrated Engineering Solution (IS)

The integration of engineering solutions takes into account the entire value chain together with the entire product lifecycle. Another opportunity an organization should benefit from when using Industry 4.0 is manufacturing innovation. The new wave of innovation is generated in production due to IoT connectivity, artificial intelligence and intelligent production [66]. Thanks to innovation in production, the new Industry 4.0 production lines become ideal for designing and launching new products in smaller quantities compared to the large volume of products currently used by some organizations. Computing technology and automation in Industry 4.0 will result in customers gaining power and control [67,68]. A major competitive advantage is obtained by using manufacturing connected and flexible systems (end-to-end integration) that use a large volume of production data and performs operations according to customer demands [69,70] throughout the product lifecycle [71,72]. Due to better use of resources in the value chain of products, end-to-end integration that is focused on creating product value will result in organizational efficiency [73].

### 2.4.4. Benefits of Digital Transformation (BDT)

Through the tools offered by the digital environment (Big Data Analytics, IoT, cloud computing) to both managers and other categories of users (department managers, specialists), but also collaborations with business partners, digital transformation encourages the formation of a digital culture. This is a valuable advantage because it forces a company to improve and continue digital learning. By adopting a strategy of continuous improvement, companies can increase their agility through digital transformation, especially through the speed of introduction to a specific target market, allowing faster adaptation and innovation.

Companies that make digital transformations improve their efficiency and profitability or can increase their market share. Automating many manual tasks and integrating data across the company allows it to streamline its workflow and improve productivity. Optimizing business technology and digital technology operations can lead to cost reductions per transaction and increased sales. Reducing marketing costs can also be achieved by practicing an improved strategy for customers, so that new digital technologies create the capabilities needed to acquire and assist customers by a company. In the conditions of digital transformation by combining data from all customer interactions and pre-existing sources in a company, it can act accordingly to optimize the customer experience and expenses.

### 2.4.5. Proposed Structural Model

According to the proposed structural model (Figure 1) the research hypotheses were formulated for both Structural Equation Modeling (SEM) and the two Multigroup Analysis (MGA) associations, as follows:

**Hypothesis 1.** *Perceived competitiveness (PC) has a positive impact on the Intention to Use Industry 4.0 processes (IUP).*

**Hypothesis 2.** *Perceived opportunities (PO) has a positive impact on the Intention to Use Industry 4.0 processes (IUP).*

**Hypothesis 3.** *Perceived risk (PR) has a positive impact on Intention to Use Industry 4.0 processes (IUP).*

**Hypothesis 4.** *Perceived vertical networking solutions (VS) have a positive impact on Intention to Use Industry 4.0 solutions (IUS).*

**Hypothesis 5.** *Perceived horizontal integration solution (HS) has a positive impact on Intention to Use Industry 4.0 solutions (IUS).*

**Hypothesis 6.** *Perceived integrated engineering solution (IS) has a positive impact on Intention to Use Industry 4.0 solutions (IUS).*

**Hypothesis 7.** *There is a positive association between Intention to Use Industry 4.0 processes (IUP) and Benefits of digital transformation (BDT).*

**Hypothesis 8.** *There is a positive association between Intention to Use Industry 4.0 solutions (IUS) and Benefits of digital transformation (BDT).*

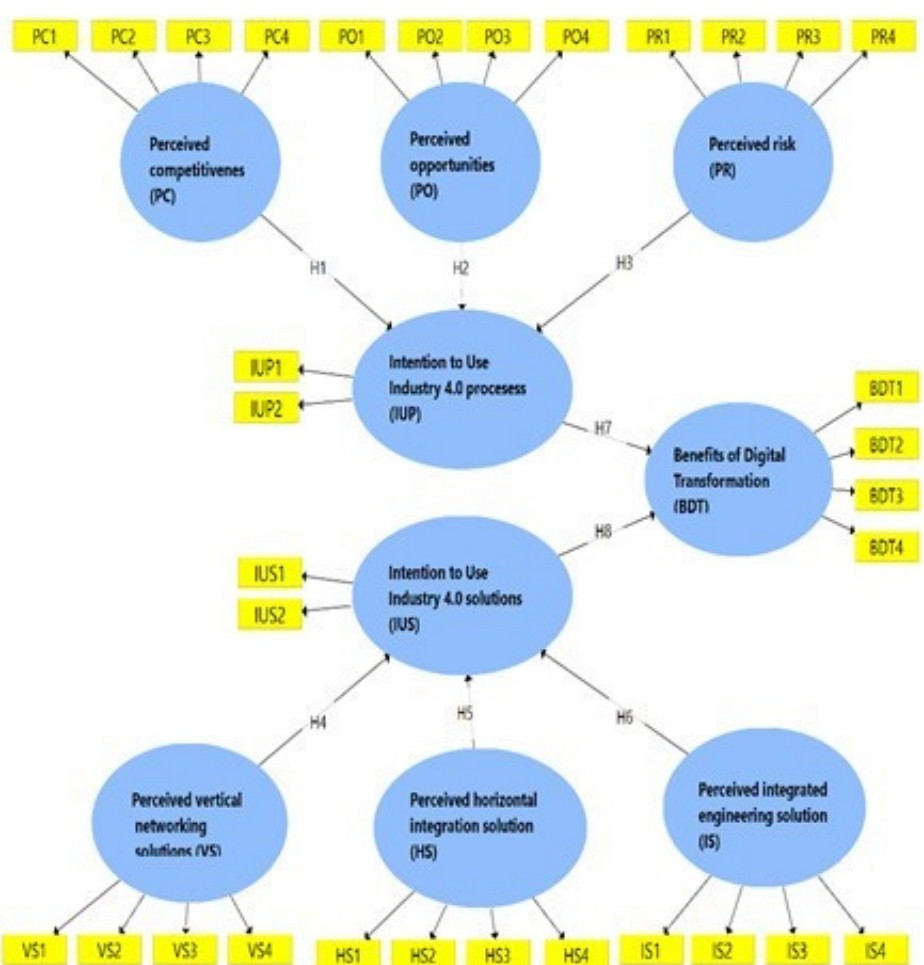

**Figure 1.** Description of the proposed structural model.

All acronyms used to describe variables and items can be found in Table A1, Appendix A.

### 3. Research Methodology

The study was based on cross-sectional marketing research, necessary to gather information on the issues investigated from a representative sample of respondents. Given the purpose and objectives of the research, the necessary sources of information were identified within the organizations. The target population was represented by Romanian companies and the sampling unit by organizational management.

### 3.1. Data Collection

The method of collecting the necessary data was based on an online survey with a high degree of structuring, conducted between December 2020 and January 2021. Being cross-sectional marketing research, the information was collected using an independent sample of respondents, only once. The online survey was preferred for its many advantages: speed of research, large sample size, low costs, convenience for respondents, absence of interview operators etc. [74].

### 3.2. Questionnaire Design

The online survey was based on a questionnaire built on only closed questions with fixed answers. Respondents had the opportunity to choose a single answer that best suited their situation, without having to formulate their own answers. The questionnaire was structured in two parts. The first part included 32 content questions measured with the Likert scale in 7 points (1 = strongly disagree, 7 = strongly agree). Appendix A Table A1 presents the exogenous and endogenous latent articles and variables of the study transposed by questions. The second part included questions that addressed the sociodemographic characteristics of the sample (company size, field of activity, use of Industry 4.0 processes and solutions). These were measured with nominal scales with two or more response variants.

### 3.3. Sampling Method and Sample Size

The sampling method was nonprobabilistic, based on the quotas identified in the target population, namely the share of companies in the domestic market. The sample extracted was representative, respecting exactly the same proportions of the community researched. The sampling base or list of companies included demographic reference variables that were provided by the National Institute of Statistics and the National Office of the Trade Register. The structure of the sample is shown in Table 1.

**Table 1.** Sample structure.

| Characteristics | N | % |
|---|---|---|
| Number of Employees | | |
| Less than 10 employees | 147 | 42.36 |
| Between 10 and 49 employees | 64 | 18.44 |
| Between 50 and 249 employees | 83 | 23.92 |
| Over 250 employees | 53 | 15.27 |
| Field of activity | | |
| Industry | 122 | 35.16 |
| Trade | 99 | 28.53 |
| Construction | 49 | 14.12 |
| Services | 77 | 22.19 |
| Do you use Industry 4.0 processes? | | |
| No | 297 | 85.59 |
| Yes | 50 | 14.41 |
| Do you use Industry 4.0 solutions? | | |
| Yes | 310 | 89.34 |
| No | 37 | 10.66 |
| Total | 347 | 100.00 |

The sample size was determined by multiplying the number of indicators by 5–10, the method advanced by [75]. Consequently, the smallest variant of the sample should have been at least 32 indicators × 5 = 160 respondents. Although over 540 questionnaires were distributed to the respondents, in the end 381 were collected, of which only 347 were valid [76]. The response rate of the respondents was 64.25%. The share of large enterprises in the surveyed sample was 15.27%. Most of them carry out activities in sectors such as industry (35.16%), trade (28.53%), services (22.19%) and others.

### 3.4. Data Processing

The data collected from the respondent were processed using SPSS and SmartPLS 3 software. All research results were obtained using the Partial Least Square–Path Model (PLS–PM), Bootstrapping and Blindfolding algorithm, and the differences between the analyzed groups were calculated using the Multigroup Analysis technique to test for group difference (PLS–MGA) [77].

## 4. Results

### 4.1. PLS–SEM Model

The PLS–SEM (Structural Equation Modeling) model is a path model that demonstrates the effect of exogenous variables on endogenous ones given the sequence of causal hypotheses. PLS regression models are frequently used for econometric modeling of growth and are the subject of several studies [78,79]. Figure 2 shows the results obtained after applying PLS–SEM modeling.

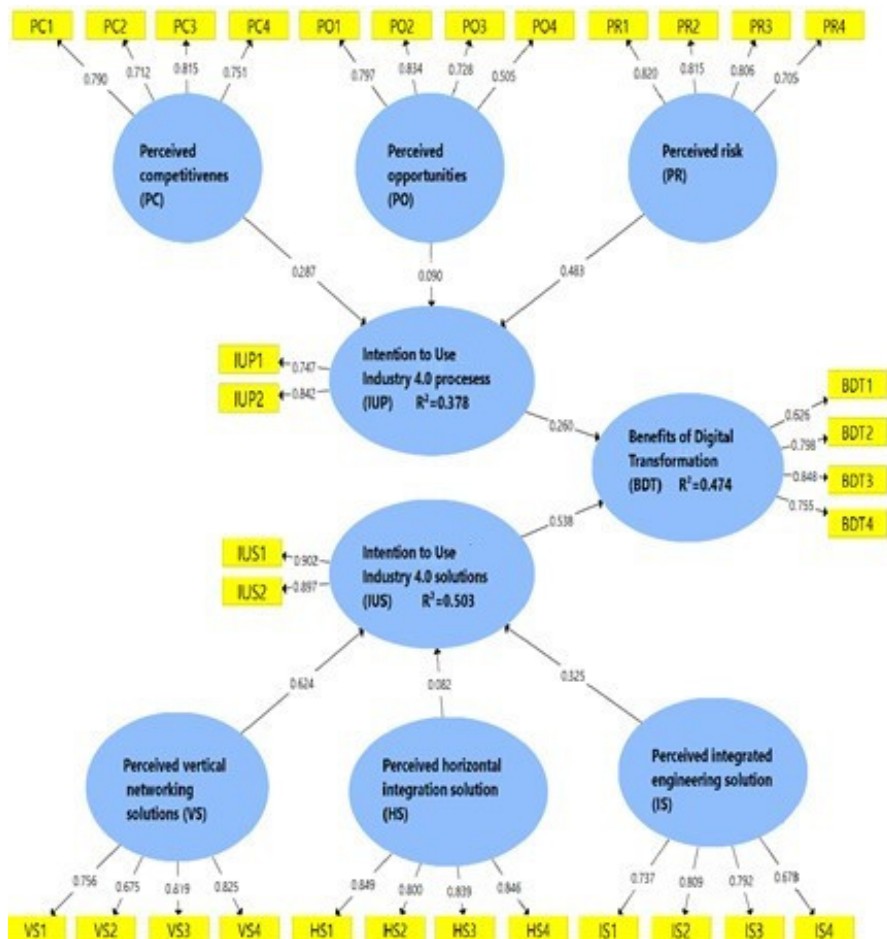

**Figure 2.** Description of the results obtained after applying Partial Least Square–Structural Equation Modeling (PLS–SEM).

### 4.2. Measurement Model

Convergent Validity

In the case of the proposed reflective model, convergent validity testing was performed using the composite reliability indicator (CR > 0.7) [80], preferred over Cronbach's alpha, which may overestimate or underestimate scale reliability [81]. The reliability of the composite reliability indicator (CR) varies between 0.775 and 0.901, being higher than the minimum allowed limit of 0.700; therefore, the estimated reliability for the proposed structural model is perfect. Although it is a conservative measure that tends to underestimate reliability, Cronbach's alpha nevertheless highlights whether indicators for latent variables display convergent validity and obvious reliability. Therefore, in the proposed reflective model, Cronbach's alpha values are between 0.700 and 0.853, demonstrating good reliability (CR > 0.7) for a good scale [82].

Although the Average Variance Extracted (AVE) reflects the average communality for each exogenous and endogenous variable of the model, it can also be used as a test of convergent and divergent validity. In the structural model, AVE varies from 0.529 to 0.809, being higher than 0.5 (Table 2) [80,83]. Therefore, the convergent and divergent validity of the proposed reflective model were demonstrated.

**Table 2.** Construction of the reliability and validity of the reflective model.

| Variables | Cronbach's Alpha (CA) | Composite Reliability (CR) | Average Variance Extracted (AVE) |
|---|---|---|---|
| BDT | 0.754 | 0.845 | 0.580 |
| HS | 0.853 | 0.901 | 0.695 |
| IS | 0.747 | 0.841 | 0.571 |
| IUP | 0.724 | 0.775 | 0.633 |
| IUS | 0.764 | 0.895 | 0.809 |
| PC | 0.771 | 0.852 | 0.590 |
| PO | 0.700 | 0.813 | 0.529 |
| PR | 0.795 | 0.867 | 0.621 |
| VS | 0.775 | 0.854 | 0.595 |

### 4.3. Discriminant Validity

The testing of the discriminant validity, by the Fornell–Larcker criterion for the latent variables of the reflective model, was based on the AVE square roots whose size was larger than their correlations with any other latent variable.

As seen in Table 3, the AVE square root of the latent variable BDT, shared with its indicator block (0.761), was superior to the variance it shares with the other latent variables: HS (0.622), IS (0.583), IUP (0.485), IUS (0.647), PC (0.431), PO (0.556), PR (0.599) and VS (0.695).

The Standardized Root Mean Square Residual (SRMR) reflects the measure of matching of the structural model, determining the difference between the observed and the implicit correlation matrix. Otherwise, we can say that the proposed model fits well, because the SRMR is 0.078, below the maximum allowable threshold of 0.08 [84].

Discriminant validity can also be determined using the Heterotrait–Monotrait Ratio (HTMT), which indicates that the difference between heterotrait and monotrait correlations should be less than 0.85 [85]. As seen in Figure A1 (Appendix A) the differences between the heterotrait and monotrait correlations of the latent variables of the model are between a minimum of 0.532 and a maximum of 0.843 (<0.85).

**Table 3.** Fornell–Larcker criterion.

|       | BDT   | HS    | IS    | IUP   | IUS   | PC    | PO    | PR    | VS    |
|-------|-------|-------|-------|-------|-------|-------|-------|-------|-------|
| BDT   | 0.761 |       |       |       |       |       |       |       |       |
| HS    | 0.622 | 0.833 |       |       |       |       |       |       |       |
| IS    | 0.583 | 0.710 | 0.756 |       |       |       |       |       |       |
| IUP   | 0.485 | 0.547 | 0.637 | 0.796 |       |       |       |       |       |
| IUS   | 0.647 | 0.589 | 0.548 | 0.419 | 0.900 |       |       |       |       |
| PC    | 0.431 | 0.612 | 0.511 | 0.448 | 0.543 | 0.768 |       |       |       |
| PO    | 0.556 | 0.690 | 0.633 | 0.525 | 0.558 | 0.707 | 0.727 |       |       |
| PR    | 0.599 | 0.710 | 0.660 | 0.604 | 0.572 | 0.604 | 0.716 | 0.788 |       |
| VS    | 0.695 | 0.718 | 0.744 | 0.606 | 0.707 | 0.632 | 0.699 | 0.749 | 0.771 |

### 4.4. Structural Model

The histograms (Figure 3a,b) below indicate the distribution of path loading coefficients for the path coefficients from IUS to BDT and from IUP to BDT, respectively.

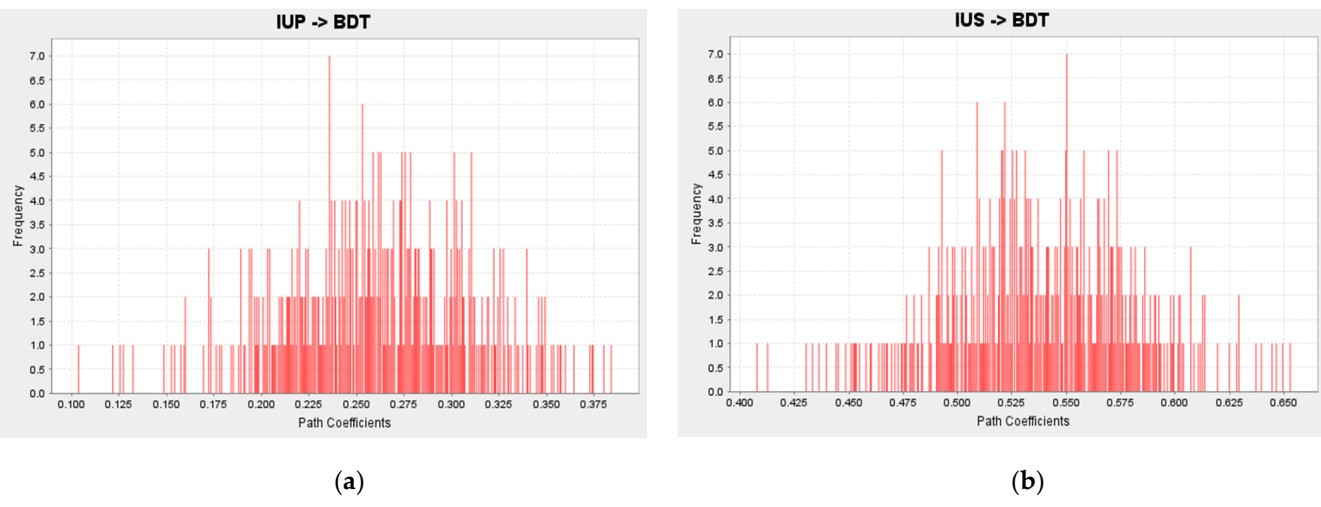

(**a**)                                              (**b**)

**Figure 3.** (**a**) Path coefficients IUP–BDT. (**b**) Path coefficients IUS–BDT.

The R-square ($R^2$) and Adjusted $R^2$ coefficients were used to measure the overall effect size of the structural model [83]. The endogenous variable IUP indicates an R-square ($R^2$) of 0.378, which means that approximately 38% of its variance is explained by the common action of PC, PO and PR factors. R-square obtained by the endogenous variable IUS was 0.503, its variance explained by the common action of VS, HS, TES. Finally, the actions of the endogenous variables IUP and IUS explain the 47.4% variance of the endogenous factor BDT (Table 4).

**Table 4.** Description of R square ($R^2$) and Adjusted $R^2$ values.

|       | R Square ($R^2$) | Adjusted $R^2$ | Stone–Gleisser ($O^2$) |
|-------|------------------|----------------|------------------------|
| BDT   | 0.474            | 0.471          | 0.254                  |
| IUP   | 0.378            | 0.372          | 0.231                  |
| IUS   | 0.503            | 0.498          | 0.397                  |

For the three endogenous factors, the cross-validated redundancies were calculated, i.e., the Stone–Gleisser value ($O^2$). As all $O^2$ values (BDT, $O^2 = 0.254$; IUP, $O^2 = 0.231$ and IUS, $O^2 = 0.397$) were greater than 0, it follows that the model is relevant for their prediction [81]. By running the bootstrapping option in the SmartPLS3 software, values for the *t*-value tests and for the probability levels (*p*-value) were generated for all paths of the structural model. The model paths are significant at *t*-value > 1.96 or *p*-value > 0.001 levels. In the case of the proposed structural model, the correlations between the variables PO -> IUP and HS -> IUS indicate *t*-value below the allowed limit of 1.96. The remaining values are significant at 0.05. The calculation results indicate four probability levels of 0.000 (PR -> IUP, VS -> IUS, IUP -> BDT and IUS -> BDT). Therefore, all four paths are significant at a probability level better than 0.001 (Table 5).

**Table 5.** Results.

| Hypotheses | Correlations | Path Coefficients | *t*-Value | *p*-Value |
|:---:|:---:|:---:|:---:|:---:|
| $H_1$ | PC -> IUP | 0.287 | 5.790 | 0.008 |
| $H_2$ | PO -> IUP | 0.090 | 0.978 | 0.328 |
| $H_3$ | PR -> IUP | 0.483 | 7.293 | 0.000 |
| $H_4$ | VS -> IUS | 0.624 | 9.270 | 0.000 |
| $H_5$ | HS -> IUS | 0.082 | 1.105 | 0.270 |
| $H_6$ | IS -> IUS | 0.325 | 6.367 | 0.014 |
| $H_7$ | IUP -> BDT | 0.260 | 5.591 | 0.000 |
| $H_8$ | IUS -> BDT | 0.538 | 13.355 | 0.000 |

*4.5. Multigroup Analysis*

PLS Multigroup Analysis (MGA) was used to test whether the PLS model differs significantly between groups for the proposed and measured variables. Initially, the model was tested to see if there were any differences between "No, I don't use Industry 4.0 processes" and "Yes, I use Industry 4.0 processes", considering that "intention to use Industry 4.0 processes" is a variable that deserves to be measured. Similarly, significant differences between the "No, I don't use Industry 4.0 solutions" and "Yes, I use Industry 4.0 solutions" groups were highlighted. To compare paths between groups, this multigroup parametric analysis uses independent samples of *t* (*t*-value) tests [86].

In order to determine whether the standardized track coefficients in the proposed (internal) structural model are higher for companies that "No, I don't use Industry 4.0 processes" than for those that use them, and to identify whether there is a difference, the t bootstrap test was used in the SmartPLS3 application. The results indicate that for the pathway from IUP to BDT, the confidence intervals overlap (No, from 0.121 lower to 0.310 higher; Yes, 0.130 lower to 0.702 higher). Similarly, the confidence intervals for the paths overlap: PC -> IUP, PR -> IUP, VS -> IUS, IS -> IUS, IUS -> BDT. Exceptions are the confidence intervals for the PO -> IUP and HS -> IUS paths, which do not overlap.

The nonparametric PLS–MGA significance test finds a significant difference between the path coefficients of the groups if the value $p < 0.05$ or $p > 0.95$ [87]. Although this test is the most commonly used, two more tests were performed, namely the parametric test, to see if the differences between the groups have equal or unequal variances using the Welch–Satterthwait Test (WST) [88]. For the proposed structural model, the results shown in Table 6 indicate that where there are differences depending on the "intention to use Industry 4.0 processes", they are not significant for these path coefficients.

The structural model was then tested to see if there were any differences between "No, I don't use Industry 4.0 solutions" and "Yes, I use Industry 4.0 solutions". After applying the t bootstrap test for the new groups analyzed as a result for the pathway from IUS to BDT, the confidence intervals overlap (No, from 0.444 lower to 0.624 higher; Yes, 0.213 lower to 0.753 higher). The confidence intervals for the paths also overlap: PC -> IUP,

VS -> IUS, IS -> IUS, IUP -> BDT. The confidence intervals for the PO -> IUP, PR -> IUP and HS -> IUS pathways do not overlap at a significance level of 0.05.

**Table 6.** Differences between groups "No, I don't use Industry 4.0 processes" and "Yes, I do use Industry 4.0 processes".

| Hypothesis | Pathways | Path Coefficients–Diff (No–Yes) | PLS–MGA | Parametric Test | | Welch–Satterthwait Test | |
|---|---|---|---|---|---|---|---|
| | | | No, I Don't Use Industry 4.0 Processes—Yes, I Do Use Industry 4.0 Processes | | | | |
| | | | *p*-Value New (No vs. Yes) | *t*-Value (\|No vs. Yes\|) | *p*-Value (No vs. Yes) | *t*-Value (\|No vs. Yes\|) | *p*-Value (No vs. Yes) |
| 1 | PC -> IUP | −0.136 | 0.404 | 0.693 | 0.489 | 0.790 | 0.433 |
| 2 | PO -> IUP | 0.203 | 0.362 | 0.808 | 0.419 | 0.902 | 0.371 |
| 3 | PR -> IUP | −0.113 | 0.485 | 0.560 | 0.576 | 0.691 | 0.492 |
| 4 | VS -> IUS | 0.161 | 0.497 | 0.838 | 0.402 | 0.678 | 0.501 |
| 5 | HS -> IUS | −0.156 | 0.477 | 0.761 | 0.447 | 0.666 | 0.508 |
| 6 | IS -> IUS | 0.025 | 0.895 | 0.136 | 0.892 | 0.126 | 0.900 |
| 7 | IUP -> BDT | −0.251 | 0.095 | 1.884 | 0.060 | 1.693 | 0.096 |
| 8 | IUS -> BDT | 0.206 | 0.172 | 1.852 | 0.065 | 1.314 | 0.195 |

The resumption of these three tests in the case of the groups "No, I don't use Industry 4.0 solutions" and "Yes, I use Industry 4.0 solutions" indicates that for the proposed structural model, there is a significant difference depending on the "intention to use Industry 4.0 solutions" for the path between PR and IUP variables (*p*-value new = 0.981> 0.95 (MGA), *p*-value new = 0.969 > 0.95 (PT) and *p*-value new = 0.977 > 0.95 (WST) (Table 7).

**Table 7.** Differences between groups "No, I don't use Industry 4.0 solutions" and "Yes, I do use Industry 4.0 solutions".

| Hypothesis | Pathways | Path Coefficients-Diff (No–Yes) | PLS–MGA | Parametric Test (PT) | | Welch–Satterthwait Test (WST) | |
|---|---|---|---|---|---|---|---|
| | | | No, I Don't Use Industry 4.0 Solutions—Yes, I Do Use Industry 4.0 Solutions | | | | |
| | | | *p*-Value New (No vs. Yes) | *t*-Value (\|No vs. Yes\|) | *p*-Value (No vs. Yes) | *t*-Value (\|No vs. Yes\|) | *p*-Value (No vs. Yes) |
| 1 | PC -> IUP | −0.295 | 0.158 | 1.276 | 0.203 | 1.178 | 0.246 |
| 2 | PO -> IUP | 0.474 | 0.166 | 1.643 | 0.101 | 1.422 | 0.163 |
| 3 | PR -> IUP | −0.009 | **0.981** | 0.039 | 0.969 | 0.029 | 0.977 |
| 4 | VS -> IUS | 0.363 | 0.091 | 1.679 | 0.094 | 1.524 | 0.136 |
| 5 | HS -> IUS | −0.257 | 0.445 | 1.044 | 0.297 | 0.811 | 0.422 |
| 6 | IS -> IUS | −0.097 | 0.649 | 0.448 | 0.655 | 0.432 | 0.668 |
| 7 | IUP -> BDT | 0.095 | 0.601 | 0.618 | 0.537 | 0.514 | 0.611 |
| 8 | IUS -> BDT | −0.024 | 0.796 | 0.178 | 0.859 | 0.180 | 0.858 |

## 5. Discussions and Conclusions

The scientific contribution of this study lies in the fact that future specialists can identify several independent and dependent latent variables, and can develop new models that contribute to the integration of digital transformation and the development of Industry 4.0 in emerging economies.

The results of the study indicate that the proposed structural model for assessing the impact of distinct factors on the intention to use Industry 4.0 processes and solutions and the benefits of digital transformation at the company level is valid and robust. Approximately 47.4% of the variance of the endogenous variable BDT is explained by the common action of IUP and IUS factors. In turn, the endogenous variables IUP and IUS indicate an R-square ($R^2$) of 0.378 and 0.503, respectively, proving once again that the structural model is strong. Given the uniqueness of the model, the results obtained will be discussed only from the perspective of working hypotheses and tests performed.

### 5.1. Theoretical Implications

According to the research results, six correlations are significant because their *t*-value levels are higher than the allowed value of 1.96, as follow: PC -> IUP (*t*-value = 5.790 > 1.96); PR -> IUP (*t*-value = 7.293 > 1.96), VS -> IUS (*t*-value = 9.270 > 1.96), IS -> IUS (*t*-value = 6.367 > 1.96), IUP -> BDT (*t*-value = 5.591 > 1.96) and IUS -> BDT (*t*-value = 13.355 > 1.96). Therefore, the null hypothesis 1, hypothesis 3, hypothesis 4, hypothesis 6, hypothesis 7, hypothesis 8 were accepted. For hypothesis 2 and hypothesis 5, the null hypotheses are rejected and the alternative ones are accepted. The exogenous latent variable PO exerts a weak, positive influence (0.090) on the IUP, just as the HS factor has a low positive impact on the IUS (0.082). The research results indicate that the variables Perceived competitiveness (PC), Perceived risk (PR) have a significant impact on Intention to Use Industry 4.0 processes (IUP) while Perceived vertical networking solutions (VS) and Perceived integrated engineering solution (IS) have a significant influence on the Intention to Use Industry 4.0 solutions (IUS). In conclusion, we can say that between Intention to Use Industry 4.0 solutions (IUS) and Benefits of Digital Transformation (BDT) there is a positive and significant association.

The first Multigroup Analysis PLS (MGA), performed between the groups "No, I don't use Industry 4.0 processes" and "Yes, I use Industry 4.0 processes" for the proposed structural model indicated that there were no differences for the six paths created by the variables proposed and measured. The confidence intervals overlap for the six: PC -> IUP, PR -> IUP, VS -> IUS, IS -> IUS, IUP -> BDT, IUS -> BDT, so we accept null hypothesis 1, hypothesis 3, hypothesis 4, hypothesis 6, hypothesis 7 and hypothesis 8. The situation is different in the case of pathways PO -> IUP and HS -> IUS, where the confidence intervals do not overlap; therefore, at a significance level of 0.05, the null hypotheses hypothesis 2 and hypothesis 5 and the alternative ones are accepted. The results indicate insignificant differences depending on the "intention to use Industry 4.0 processes" for these path coefficients.

The second Multigroup Analysis PLS (MGA), carried out between the groups "No, I do not use Industry 4.0 solutions" and "Yes, I use Industry 4.0 solutions" showed that there are no differences between the paths IUS -> BDT, PC -> IUP, VS -> IUS, IS -> IUS, IUP -> BDT, so the null hypotheses are accepted: hypothesis 1, hypothesis 4, hypothesis 6, hypothesis 7 and hypothesis 8. The confidence intervals for the pathways PO -> IUP, PR -> IUP and HS -> IUS do not overlap, so we can say that there are differences and we accept the alternative hypotheses hypothesis 2, hypothesis 3 and hypothesis 5. After performing the PLS–MGA, the parametric test and the Welch–Satterthwait test, it turned out that there is a significant difference depending on the "intention to use Industry 4.0 solutions" for the path between the latent variables PR and IUP.

### 5.2. Managerial Implications

Perceived competitiveness (PC) and Perceived risk (PR) have a significant influence on the organization's management intention to use Industry 4.0 processes, while Perceived opportunities (PO) has a weak but positive impact. The average appreciation of the management of the companies regarding obtaining some competitive advantages through the combined use of several strategies such as low-cost strategies (PC1), integration strategies (PC2), intensive strategies (PC3) and differentiation strategies (PC4), were at a medium level (3.5 points) with a mode of 4 points (for a scale from 1 = strongly disagree to 7 = strongly agree). The activity of companies differs from one branch to another; they face different situations due to the interaction of factors that generate cyber security (PR1), operation (PR2) technology (PR3) and data leakage (PR4) risks.

Company management confirms that it constantly implements procedures and measures to prevent unwanted internal and external events, reduce risks regarding the unauthorized access and confidentiality of technology systems, reduce losses due to technological failures and protect data across the digital ecosystem (mean = mode = 5 points). A significant impact on the intention to use Industry 4.0 solutions had the following factors:

Perceived vertical networking solutions (VS) and Perceived integrated engineering solution (IS). The variable Perceived horizontal integration solution (HS) demonstrated only a weak, positive influence. To help companies manage the transition to Industry 4.0, managers continue to adopt a wide range of solutions such as IT Integration (VS1), Analytics and data management (VS2), Cloud-based applications (VS3) and Operational efficiency 2.0. (VS4) (Median = 5 points, Mode = 6 points). Respondents agree that they must be doubled by new management solutions, among which we mention the "new types of innovation" (IS1), Efficient management of innovation (IS2), Efficient lifecycle management (IS3) and Efficient holistic management (IS4) (median = 5, mode = 5). Company management agrees that digital transformation produces multiple benefits including Improved productivity (BDT1), Increased agility (BDT2), Increased profits (BDT3) and Encourages digital culture (BDT4) (median = 5.5, mode = 7).

Given the research results, the main proposal for managing organizations is the use of Industry 4.0 processes and solutions to obtain benefits such as streamlining workflow and improving productivity, increasing the speed of marketing innovative products, improving employee work tools for good communication and performing work tasks, among others.

Compared to other specialists' studies [89–91], the research presents some limiting conditions related to the high degree of structuring of the questionnaire, the lack of detail of the answers given by the participants and the lack of diversity of scales used for questions.

Future research will overcome these limitations by identifying other latent variables that influence digital transformation, by asking open-ended questions that include respondents' answers and by measuring data through a variety of scales.

**Author Contributions:** Conceptualization, S.C. and M.C.T.; data curation, C.M.B. and M.T.F.; formal analysis, A.I.S.; investigation, A.I.S.; methodology, M.C.T.; resources, D.M. and M.C.T.; supervision, S.C.; validation, D.M., C.M.B., D.I.T. and L.S.; visualization, D.I.T. and M.T.F.; writing—original draft, M.C.T.; writing—review and editing, S.C. All authors have read and agreed to the published version of the manuscript.

**Funding:** This research received no external funding.

**Informed Consent Statement:** Informed consent was obtained from all subjects involved in the study.

**Conflicts of Interest:** The authors declare no conflict of interest.

## Appendix A

**Table A1.** Questionnaire.

| Variables | Items |
|---|---|
| Perceived competitiveness (PC) | **(PC1) Low-cost strategies** Does your organization's management agree to gain a competitive advantage by charging minimal unit costs for the delivery of equivalent products or services? |
| | **(PC2) Integration strategies** Does the management of your organization agree to gain a competitive advantage by practicing growth strategies through integration, by expanding activities downstream, upstream or at the same level within the same field of activity or related activities? |
| | **(PC3) Intensive strategies** Does your organization's management agree to gain a competitive advantage by practicing intensive growth strategies by increasing the volume of sales at the expense of new and existing products or services on the market? |
| | **(PC4) Differentiation strategies** Does your organization's management agree to gain a competitive advantage by practicing differentiation strategies when developing new products and services? |

**Table A1.** *Cont.*

| Variables | Items |
|---|---|
| Perceived opportunities (PO) | **(PO1) Integration of customer preferences**<br>Does your organization's management agree with the integration of customer preferences into the development and production process to increase quality and efficiency? |
| | **(PO2) Adjust/Improve talent**<br>Does your organization's management agree with the retraining and further training of employees to enable digital transformation before integration into Industry 4.0? |
| | **(PO3) New exponential technologies**<br>Does your organization's management agree with the use of key 3D printing technology (additive manufacturing) in production and logistics processes to accelerate digital transformation after integration into Industry 4.0? |
| | **(PO3) New business segment development**<br>Does your organization's management agree with the development of new business segments (research and development, procurement, production, warehousing and logistics) that are at the heart of digital transformation after integration into Industry 4.0? |
| Perceived risk (PR) | **(PR1) Cyber-security risk**<br>Does your organization's management agree to adopt internal procedures aimed at protecting the digital environment by blocking unauthorized access / use and ensuring the confidentiality and integrity of technology systems (e.g., platform strengthening, network architecture, security application, vulnerability management and security monitoring)? |
| | **(PR2) Operations risk**<br>Does your organization's management agree with the implementation of internal or external event prevention procedures that may adversely affect the ability to meet business objectives through its defined operations (e.g., inadequate controls in operating procedures)? |
| | **(PR3) Technology risk**<br>The management of your organization agrees with the implementation of measures to reduce the potential for losses due to technological failures or obsolete technologies, which have a major impact on systems, people and processes (examples of risk areas: scalability, file compatibility and accuracy, functionality of implemented technology )? |
| | **(PR4) Data Leakage risk**<br>Your organization's management agrees with the implementation of data protection procedures across the digital ecosystem at different stages of the data lifecycle—data in use, data in transit and data at rest (for example: control areas identification: data classification, data storage, data processing, data encryption, etc.)? |
| Intention to Use Industry 4.0 processes (IUP) | **(IUP1)** Does your organization's management intend to use Industry 4.0 processes to develop future corporate investments? |
| | **(IUP2)** Does your organization's management intend to use Industry 4.0 processes to create new business areas that can become future business centers? |
| Perceived vertical networking solutions (VS) | **(VS1) IT Integration**<br>Does your organization's management agree with the implementation of the latest IT solutions (e.g., sensors, modules, control systems, communications networks, business applications, etc.) to maintain its long-term market advantage? |
| | **(VS2) Analytics and data management**<br>Does your organization's management agree with the development of specific skills in the areas of analytics and efficient big data management and the integration of new business processes based on the information provided by specific analyzes? |
| | **(VS3) Cloud-based applications**<br>Does your organization's management agree with the implementation of cloud-based solution networks for hosting and efficient use of big data generated by integration into Industry 4.0? |

**Table A1.** *Cont.*

| Variables | Items |
|---|---|
| Perceived vertical networking solutions (VS) | **(VS4) Operational efficiency 2.0.**<br>Does your organization's management agree with the integration of specific processes for the analysis, evaluation and efficient application of data collected from equipment / sensors for making the fastest decisions on safety, work processes, maintenance and service? |
| Perceived horizontal integration solution (HS) | **(HS1) Business model optimization**<br>Does the management of your organization agree with the implementation of a radically different new business model than with simply improving the established one? |
| | **(HS2) Smart supply chains**<br>Does your organization's management agree with the implementation of new, smarter, more transparent and more efficient supply chains, adapted to customer needs and allowing new forms of cooperation with business partners? |
| | **(HS3) Smart logistics**<br>Does your organization's management agree with the implementation of intelligent, autonomous, flexible logistics processes, connection to the new generations of global value chain networks? |
| | **(HS4) IT security management**<br>Does your organization's management agree with the integration of a risk management system and a cyber-security strategy to improve operational security and protection against attacks along the value chain? |
| Perceived integrated engineering solution (IS) | **(IS1) The "new types of innovation"**<br>The management of your organization agrees to implement new Industry 4.0 solutions in addition to the traditional ones (for example: innovations related to goods offers, changing processes, networks and profit models, new distribution channels, new uses for a strong brand, etc.)? |
| | **(IS2) Efficient management of innovation**<br>Does your organization's management agree to implement innovation management solutions that accelerate research and development, track innovation return on investment (ROI), identify risks through the use of global comparative data, etc.? |
| | **(IS3) Efficient lifecycle management**<br>Does your organization's management agree to implement efficient lifecycle management solutions that enable the collection and processing of big data, the generation of specific early indicators through the use of artificial intelligence (AI) and the creation of relevant bases for future decision making? |
| | **(IS4) Efficient holistic management**<br>Does your organization's management agree to implement effective holistic management solutions that enable the best decisions to be made economically, ecologically and socially and to develop a clear vision of the desired future in the field of digital transformation? |
| Intention to Use Industry 4.0 solutions (IUS) | **(IUS1)** Does your organization's management intend to use Industry 4.0 solutions to implement new business management solutions that promote and support innovation, involvement and reward? |
| | **(IUS1)** Does your organization's management intend to use Industry 4.0 solutions to create and develop successful business processes and segments to strengthen its market leadership? |
| Benefits of Digital Transformation (BDT) | **(BDT1) Improved productivity**<br>Does your organization's management agree that the right technology tools can automate many manual tasks and integrate data across departments, streamline workflow, and improve productivity? |

**Table A1.** *Cont.*

| Variables | Items |
|---|---|
| Benefits of Digital Transformation (BDT) | **(BDT2) Increased agility**<br>Does your management agree that the digital transformation makes the organization more agile by increasing the speed of marketing innovative products and services and by adopting continuous improvement strategies? |
| | **(BDT3) Increased profits**<br>Does your management agree that the digital transformation makes the organization more efficient and profitable by increasing the volume of its revenue faster than that of the competitors? |
| | **(BDT4) Encourages digital culture**<br>Does your organization's management agree that digital transformation encourages digital culture by providing employees with the right tools, tailored to their environment, for good collaboration, communication, and easy work? |

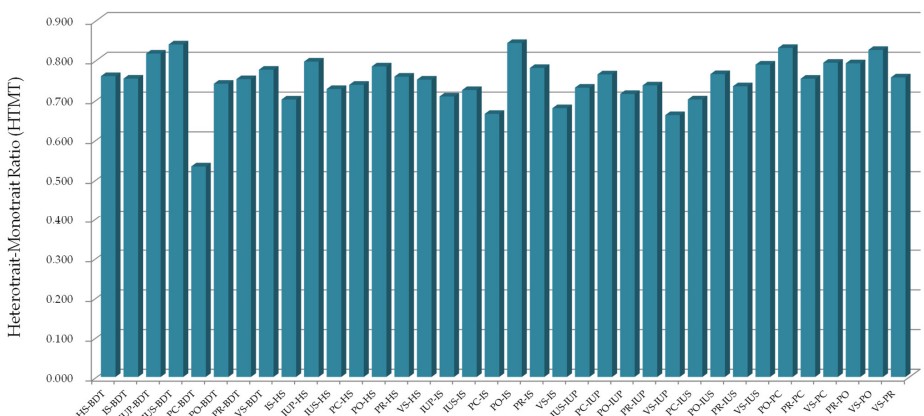

**Figure A1.** Heterotrait–Monotrait Ratio (HTMT).

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
