# Peer review of "The Impact of Force Factors on the Benefits of Digital Transformation in Romania"

_applsci, doi:10.3390/app11052365_

Round 1

Reviewer 1 Report

The paper aims at discovering which factors are significant for the formulation of the behavioral intention to adopt Industry 4.0 processes and solutions by Romanian managers. Overall, the implemented statistical methodology is robust, and the internal validity seems high. The model is scientifically sound. The topics treated are relevant and may catch the attention even of a broader public. Altogether, the paper is worthy of publication. Minor improvements are, however, possible (and therefore needed).

Title

The title does not seem to capture the principal goal of the paper, which is, according to the Abstract, determining which forces foster the adoption of Industry 4.0 processes and solutions the most. In its present form, the title is too generic. It does not suit the methodology-heavy content of the paper.

Keywords

The last three keywords (challenges, solutions, benefits) are very generic. If possible, it is better to change them or get rid of them altogether, as they are superfluous and may suit whatever paper on whatever topic.

Introduction

The Introduction section is well-written and concise, so there is no reason to repeat the sentence already used in the Abstract (in Italics, see lines 54-56). It would be better to reformulate the sentence to avoid unnecessary repetition.

Materials and Methods

Because of the required citation style, it is better to change the way items and variables are listed (see lines 76-87 and lines 140-141). The numbers denoting citations and the numbers introducing variables intermix and hinder the clarity of the exposure. A different way of listing items is advised (for example, using letters instead of numbers).

As for the content, it is not clear how the UTAUT four exogenous variables relate to the three variables that inform the model (IUP, IUS, and BDT). Only one short paragraph is devoted to the issue (lines 133-148). However, being such a critical point, it requires a more detailed explanation.

The set of hypotheses as reported at the end of the section is semantically ambiguous. For example, the third hypothesis: perceived risk has a positive impact on IUP. But what kind of it is intended for the relationship? The higher perceived risk or lower perceived risk? The same is true for the rest of the hypotheses. It is better to specify the degree for each variable (higher PC, high PO, etc...) If not, the hypothesized connection between the variables could be easily misinterpreted.

Research methodology

This section presents robust statistical modeling that seems impeccable from the points of view of both internal validity and reliability. Nonetheless, the abundance of figures and tables that elaborate on the model's assumptions but are not significant for the final discussion unnecessarily complicates the exposure. The section feels crowded and therefore less "digestible." Some figures and tables (for example Figure 3) can be moved to the Appendix section without undermining the clarity and the quality of the exposure.

Discussions and conclusions

Instead of elaborating on the results of the model, the authors report here what can be easily extrapolated from the graphs and tables of the previous section. "Discussions and conclusions" should be enriched with more in-depth considerations and implications for managers while some of the more technical examination of the results better suits the "Research methodology" section.

Author Response

Dear Reviewer,

Thank you for your comments. Appended to this letter is our point-by-point response to the comments. The comments are reproduced and our responses are given directly afterward in a different color (red).

The paper aims at discovering which factors are significant for the formulation of the behavioral intention to adopt Industry 4.0 processes and solutions by Romanian managers. Overall, the implemented statistical methodology is robust, and the internal validity seems high. The model is scientifically sound. The topics treated are relevant and may catch the attention even of a broader public. Altogether, the paper is worthy of publication. Minor improvements are, however, possible (and therefore needed).

Title

The title does not seem to capture the principal goal of the paper, which is, according to the Abstract, determining which forces foster the adoption of Industry 4.0 processes and solutions the most. In its present form, the title is too generic. It does not suit the methodology-heavy content of the paper.

Response: In order to capture the main purpose of the paper and according to the reviewer's suggestion, the initial title of article "The Benefits of Digital Transformation in the Romanian Business Environment" was replaced with the new title "The Impact of Force Factors on the Benefits of Digital Transformation in Romania".

Keywords

The last three keywords (challenges, solutions, benefits) are very generic. If possible, it is better to change them or get rid of them altogether, as they are superfluous and may suit whatever paper on whatever topic.

Response: The three keywords indicated have been removed and replaced with the following keyword: business environment.

Introduction

The Introduction section is well-written and concise, so there is no reason to repeat the sentence already used in the Abstract (in Italics, see lines 54-56). It would be better to reformulate the sentence to avoid unnecessary repetition.

Response: In the abstract the following sentence ”Our research started with the question: What are the forces that influence the intention to use processes and solutions in Industry 4.0 and radically change the business models of organizations, generating new challenges and benefits?” was replaced by the following sentence ”There are many factors that can influence the intention to use Industry 4.0 processes and solutions and change the behavior of organizations and their business models” (lines 21-22).

Materials and Methods

Because of the required citation style, it is better to change the way items and variables are listed (see lines 76-87 and lines 140-141). The numbers denoting citations and the numbers introducing variables intermix and hinder the clarity of the exposure. A different way of listing items is advised (for example, using letters instead of numbers).

Response: According to the reviewer's suggestion, the numbers on lines 92-103, 133-135 and 167-168 have been replaced by letters.

As for the content, it is not clear how the UTAUT four exogenous variables relate to the three variables that inform the model (IUP, IUS, and BDT). Only one short paragraph is devoted to the issue (lines 133-148). However, being such a critical point, it requires a more detailed explanation.

Response: The following sentence was introduced on lines 178-180: The relationship between the four exogenous variables of the UTAUT model and the three variables that inform the proposed model (IUP, IUS and BDT) are described below. It explains what is to be explained and is presented below. All the necessary explanations required by the reviewer are presented in 2.2.1 to 2.2.7.

The set of hypotheses as reported at the end of the section is semantically ambiguous. For example, the third hypothesis: perceived risk has a positive impact on IUP. But what kind of it is intended for the relationship? The higher perceived risk or lower perceived risk? The same is true for the rest of the hypotheses. It is better to specify the degree for each variable (higher PC, high PO, etc...) If not, the hypothesized connection between the variables could be easily misinterpreted.

Response: The Perceived risk variable includes the following risk categories:

(PR1) Cybersecurity risk. The management of your organization agrees to adopt internal procedures aimed at protecting the digital environment by blocking unauthorized access / use and ensuring the confidentiality and integrity of technology systems (e.g.: platform strengthening, network architecture, security application, vulnerability management and security monitoring)?

(PR2) Operations risk. The management of your organization agrees with the implementation of event prevention procedures, internal or external, which may adversely affect the ability to meet business objectives through its defined operations (for example: inadequate controls in operating procedures)?

(PR3) Technology risk. The management of your organization agrees with the implementation of measures to reduce the potential for losses due to technological failures or obsolete technologies, which have a major impact on systems, people and processes (examples of risk areas: scalability, compatibility and accuracy file, the functionality of the implemented technology)?

(PR4) Data Leakage risk. Your organization's management agrees with the implementation of data protection procedures across the digital ecosystem at different stages of the data life cycle - data in use, data in transit and data at rest (for example: identification control: data classification, data storage, data processing, data encryption etc.)?

The perceived risk barrier was not assessed. The situation is similar for the other variables analyzed.

Research methodology

This section presents robust statistical modeling that seems impeccable from the points of view of both internal validity and reliability. Nonetheless, the abundance of figures and tables that elaborate on the model's assumptions but are not significant for the final discussion unnecessarily complicates the exposure. The section feels crowded and therefore less "digestible." Some figures and tables (for example Figure 3) can be moved to the Appendix section without undermining the clarity and the quality of the exposure.

Response: According to the reviewer request, Figure 3 has been relocated to Appendix A. To line 413 was added: ”(Appendix A)”.

Discussions and conclusions

Instead of elaborating on the results of the model, the authors report here what can be easily extrapolated from the graphs and tables of the previous section. "Discussions and conclusions" should be enriched with more in-depth considerations and implications for managers while some of the more technical examination of the results better suits the "Research methodology" section.

Response: The following sentence was introduced in the Conclusions section (lines 568-572): ”Given the research results, the main proposal for managing organizations is the use of Industry 4.0 processes and solutions to obtain benefits such as: streamlining workflow and improving productivity, increasing the speed of marketing innovative products, improving employee work tools for good communication and for performing work tasks and others”.

Reviewer 2 Report

The paper covers the current subject of implementing solutions of the "Industry 4.0" concept in the activities of enterprises.  The authors conducted research on a sample of nearly 350 units, so the formulated results can be considered reliable.

 The article is interesting, well-written, but needs some corrections (if possible) before publication:

 - please separate "Literature review" and "Methodology" sections in the paper,

 - with the proposed model (Fig.1), I suggest explaining what the elements are: PC1…;  PO1…., PR1….  etc.  This will make it easier for the reader to understand the authors' intention,

 - in the "Conslusions" section there should be a reference to studies conducted and published by other authors and an indication of the limitations accompanying the conducted studies, as well as an indication of the direction of further research.  In this part, the authors can use the following publications:

Ślusarczyk, B., Tvaronavičienė, M., Haque, A. U., & Judit, O. L. Á. H. (2020). Predictors of Industry 4.0 technologies affecting logistic enterprises’ performance: international perspective from economic lens. Technological and economic development of economy26(6), 1263-1283.

Afonasova, M. A., Panfilova, E. E., Galichkina, M. A., & Ślusarczyk, B. (2019). Digitalization in economy and innovation: The effect on social and economic processes. Polish Journal of Management Studies19.

Ślusarczyk, B., & UL HAQUE, A. (2019). Public services for business environment: challenges for implementing Industry 4.0 in Polish and Canadian logistic enterprises. Administration & Public Management Review, (33).

Author Response

Dear Reviewer,

Thank you for your comments. Appended to this letter is our point-by-point response to the comments. The comments are reproduced and our responses are given directly afterward in a different color (red).

The paper covers the current subject of implementing solutions of the "Industry 4.0" concept in the activities of enterprises.  The authors conducted research on a sample of nearly 350 units, so the formulated results can be considered reliable.

 The article is interesting, well-written, but needs some corrections (if possible) before publication:

 - please separate "Literature review" and "Methodology" sections in the paper,

Response: The title of the section "Materials and methods" was changed to "2. Literature review" keeping the numbering in the article.

 - with the proposed model (Fig.1), I suggest explaining what the elements are: PC1…;  PO1…., PR1….  etc.  This will make it easier for the reader to understand the authors' intention,

Response: Under Figure 1, line 322 the following sentence has been introduced: ”All acronyms used to describe variables and items can be found in Table 1, Appendix A”.

 - in the "Conslusions" section there should be a reference to studies conducted and published by other authors and an indication of the limitations accompanying the conducted studies, as well as an indication of the direction of further research.  In this part, the authors can use the following publications:

Ślusarczyk, B., Tvaronavičienė, M., Haque, A. U., & Judit, O. L. Á. H. (2020). Predictors of Industry 4.0 technologies affecting logistic enterprises’ performance: international perspective from economic lens. Technological and economic development of economy26(6), 1263-1283.

Afonasova, M. A., Panfilova, E. E., Galichkina, M. A., & Ślusarczyk, B. (2019). Digitalization in economy and innovation: The effect on social and economic processes. Polish Journal of Management Studies19.

Ślusarczyk, B., & UL Haque, A. (2019). Public services for business environment: challenges for implementing Industry 4.0 in Polish and Canadian logistic enterprises. Administration & Public Management Review, (33).

Response: In the Conclusions section, at line 573, the “Compared to other specialists’ studies [89-91]” was introduced and the three bibliographic references indicated by the reviewer at the end of the article were added.

Reviewer 3 Report

I had the pleasure of reviewing the manuscript entitled "The Benefits of Digital Transformation in the Romanian Business Environment". The article deals with the topic of digital transformation through an online survey in Romania. The aim is to validate a specific model developed by the authors. I found the article quite interesting, but I think that the authors should deepen some aspects and revise, in some parts, the structure of the paper. For this reason I suggest major revisions. In the following there are more aspects that, in my opinion, are necessary to improve the paper:

- There is a lack of explicit reference to which gaps the authors want to fill with their study. The authors should make it clear why their work is of interest to the scientific community and what elements are new compared to studies already in the literature. There are many studies in the literature similar to this one, for example: 
- Zheng, T., Ardolino, M., Bacchetti, A., Perona, M., & Zanardini, M. (2019). The impacts of Industry 4.0: a descriptive survey in the Italian manufacturing sector. Journal of Manufacturing Technology Management.
- Tortorella, G.L. and Fettermann, D. (2018), "Implementation of Industry 4.0 and lean production in Brazilian manufacturing companies", International Journal of Production Research, Vol. 56 No. 8, pp. 2975-2987.
These articles are not mentioned in the article! The authors should therefore better explain the gaps in the literature.

-I suggest splitting up the research questions. Currently there is only one research question. Perhaps it is more understandable to divide the questions into bullet points.

- This passage is not clear. Is there something missing? (line 83, page 2 - "Digital transformation is the deliberate and continuous digital of a company's evolution, ...")

- When talking about the business model (section 2.1), there is no mention of the platform model (in particular the digital multisided platform model). I suggest to give space also to this model e.g. Ardolino, M., Saccani, N., Adrodegari, F., & Perona, M. (2020). A business model framework to characterize digital multisided platforms. Journal of Open Innovation: Technology, Market, and Complexity, 6(1), 10.

- What are the technologies that characterise Industry 4.0 and what are their main applications? In my opinion, it is necessary to give space to this issue when discussing ICT technologies in section 2.1 (e.g. Zheng, T., Ardolino, M., Bacchetti, A., & Perona, M. (2020). The applications of Industry 4.0 technologies in manufacturing context: a systematic literature review. International Journal of Production Research, 1-33).

- The authors describe the variables in chapter 2.2. 
In order to improve the readability of the paper I think it could be useful to insert a table in which the variables are summarised, with a brief description and the main sources/references from which they have been taken.

- The title alone indicates that the companies surveyed are Romanian, but this must be clearly stated in the paper. Furthermore, the authors must justify this choice and specify why the results they obtained can be generalised. 

- The discussion part is too short and not very thorough. The discussion should critically analyse the results part and evaluate possible alignments or misalignments with the scientific literature. 

- Managerial implications are very scarce. In this paragraph, the authors should elaborate on their results by highlighting how practitioners can use them to organise their digitisation strategy. The current approach in paragraph 5.2 is unclear and does not specify this. Furthermore, limitations and conclusions should be placed in a separate paragraph.

Author Response

Dear Reviewer,

Thank you for your comments. Appended to this letter is our point-by-point response to the comments. The comments are reproduced and our responses are given directly afterward in a different color (red).

I had the pleasure of reviewing the manuscript entitled "The Benefits of Digital Transformation in the Romanian Business Environment". The article deals with the topic of digital transformation through an online survey in Romania. The aim is to validate a specific model developed by the authors. I found the article quite interesting, but I think that the authors should deepen some aspects and revise, in some parts, the structure of the paper. For this reason I suggest major revisions. In the following there are more aspects that, in my opinion, are necessary to improve the paper:

- There is a lack of explicit reference to which gaps the authors want to fill with their study. The authors should make it clear why their work is of interest to the scientific community and what elements are new compared to studies already in the literature. There are many studies in the literature similar to this one, for example: 
- Zheng, T., Ardolino, M., Bacchetti, A., Perona, M., & Zanardini, M. (2019). The impacts of Industry 4.0: a descriptive survey in the Italian manufacturing sector. Journal of Manufacturing Technology Management.
- Tortorella, G.L. and Fettermann, D. (2018), "Implementation of Industry 4.0 and lean production in Brazilian manufacturing companies", International Journal of Production Research, Vol. 56 No. 8, pp. 2975-2987.
These articles are not mentioned in the article! The authors should therefore better explain the gaps in the literature.

Response: At line 159, the two references sources indicated by the reviewer were introduced as existing gaps in our study; following the line 171, the word “Romanian” was added to highlight the specific character in the application of the case study compared to those mentioned by the reviewer. The explanation for “what elements are new compared to studies already in the literature” results in lines 171-176, which were already presented. In addition, the following was added to lines 176-180: ”By creating a new conceptual model based on the above-mentioned variables, our study  covers some existing gaps in the literature and opens up new opportunities for future research in both academia and business”. Also in lines 496-499 the following text was introduced: ”The scientific contribution of this study lies in the fact that future specialists can identify several independent and dependent latent variables, can develop new models that contribute to the integration of digital transformation and the development of Industry 4.0 in emerging economies”.

[37] Zheng, T., Ardolino, M., Bacchetti, A., Perona, M., & Zanardini, M. (2019). The impacts of Industry 4.0: a descriptive survey in the Italian manufacturing sector. Journal of Manufacturing Technology Management.

[38] Tortorella, G.L. and Fettermann, D. (2018), "Implementation of Industry 4.0 and lean production in Brazilian manufacturing companies", International Journal of Production Research, Vol. 56 No. 8, pp. 2975-2987.

-I suggest splitting up the research questions. Currently there is only one research question. Perhaps it is more understandable to divide the questions into bullet points.

Response: The research questions was split in two separated by the bullet points as follows: ”What are the forces that influence the intention to use processes and solutions in Industry 4.0; and how can they radically change the business models of organizations, generating new challenges and benefits?” In addition, "how can they" was added to achieve the agreement in the sentence (line 55)!

- This passage is not clear. Is there something missing? (line 83, page 2 - "Digital transformation is the deliberate and continuous digital of a company's evolution, ...")

Response: There is nothing missing. It is the idea taken by quotation from the author of source [19]!

- When talking about the business model (section 2.1), there is no mention of the platform model (in particular the digital multisided platform model). I suggest to give space also to this model e.g. Ardolino, M., Saccani, N., Adrodegari, F., & Perona, M. (2020). A business model framework to characterize digital multisided platforms. Journal of Open Innovation: Technology, Market, and Complexity, 6(1), 10.

Response: In lines 135-136 was introduced: (c) “a model framework to characterize digital multisided platforms” with reference note suggested by the reviewer [30].

[30] Ardolino, M., Saccani, N., Adrodegari, F., & Perona, M. (2020). A business model framework to characterize digital multisided platforms. Journal of Open Innovation: Technology, Market, and Complexity, 6(1), 10.

- What are the technologies that characterise Industry 4.0 and what are their main applications? In my opinion, it is necessary to give space to this issue when discussing ICT technologies in section 2.1 (e.g. Zheng, T., Ardolino, M., Bacchetti, A., & Perona, M. (2020). The applications of Industry 4.0 technologies in manufacturing context: a systematic literature review. International Journal of Production Research, 1-33).

Response: In lines 140-149 the following paragraph has been inserted with two reference notes, one is indicated by the reviewer as follows: ”The Industry 4.0 concept is a branch of material production in which innovative elements and technologies are integrated (Big Data & Analytics), various devices are implemented (cyber-physical systems-CPS, Internet of Things, cloud computing) and functional aspects are addressed as services, ensuring their constant communication and relation-ship [31]. The main applications of Industry 4.0 are: Internet of Objects (IoT), Embedded Software, Big Data and Data Analytics, Machine to Machine Communication (M2M), Cloud Solutions, Intelligent Robot Automation Systems, End-to-End Software Integration, Augmented Reality, Simulation, Additive Production (3D Printing), Cyber Security, Central Monitoring and Control (SCADA), Mobile Devices, Smart Sensors (RFID, QRC, BAR-CODE), Smart Objects, Remote control [32]”.

[31] Forbes (2018). What is Industry 4.0? Here's A Super Easy Explanation For Anyone. Retrieved from https://www.forbes.com/sites/bernardmarr/2018/09/02/what-is-industry-4-0-heres-a-super-easy-explanation-for-anyone/#673255c99788.

[32] Zheng, T., Ardolino, M., Bacchetti, A., & Perona, M. (2020). The applications of Industry 4.0 technologies in manufacturing context: a systematic literature review. International Journal of Production Research, 1-33).

- The authors describe the variables in chapter 2.2. In order to improve the readability of the paper I think it could be useful to insert a table in which the variables are summarised, with a brief description and the main sources/references from which they have been taken.

Response: In order to follow the other recommendations of the reviewers, namely the degree of loading with tables and graphs in the article, we consider that it is better not to enter your suggestion. In this way we also avoid repeating the information from the literature and repeating the bibliographic sources.

- The title alone indicates that the companies surveyed are Romanian, but this must be clearly stated in the paper. Furthermore, the authors must justify this choice and specify why the results they obtained can be generalised.

Response: Please see lines 59-60 or line 328.

- The discussion part is too short and not very thorough. The discussion should critically analyse the results part and evaluate possible alignments or misalignments with the scientific literature. 

- Managerial implications are very scarce. In this paragraph, the authors should elaborate on their results by highlighting how practitioners can use them to organise their digitisation strategy. The current approach in paragraph 5.2 is unclear and does not specify this. Furthermore, limitations and conclusions should be placed in a separate paragraph.

Response: The following phrases have been introduced as follows (Discussions and Conclusions section):

Lines 493-496: ”The scientific contribution of this study lies in the fact that future specialists can identify several independent and dependent latent variables, can develop new models that contribute to the integration of digital transformation and the development of Indus-try 4.0 in emerging economies”.

Lines 570-574: ”Given the research results, the main proposal for managing organizations is the use of Industry 4.0 processes and solutions to obtain benefits such as: streamlining workflow and improving productivity, increasing the speed of marketing innovative products, im-proving employee work tools for good communication and for performing work tasks and others”.

Reviewer 4 Report

It is a very interesting and useful manuscript.

The paper is well-organized with a good state-of-the-art. The research design is appropriate with a well-described methodology.

Finally, the figures are relevant and the conclusions are supported by the results.

Therefore, I recommend its publication in current form.

Author Response

Dear Reviewer,

Thank you for your comments. Appended to this letter is our point-by-point response to the comments. The comments are reproduced and our responses are given directly afterward in a different color (red).

It is a very interesting and useful manuscript.

The paper is well-organized with a good state-of-the-art. The research design is appropriate with a well-described methodology.

Finally, the figures are relevant and the conclusions are supported by the results.

Therefore, I recommend its publication in current form.

Response: On behalf of all the authors, thank you for your appreciation and recommendation.

Reviewer 5 Report

This is example of very important and highly-interesting study. The paper itself is informative and stimulating, it is well-written and well-structured. The number of sources is enough. The figures and tables are proper. I think the paper needs only small amendments, after which it can be accepted.

  • Abstract: please, avoid abbreviations and tell a bit more about your findings.
  • Introduction: please, state that your study focuses on Romania and explain why.
  • Section 2: you need to add a subsection characterizing the Romanian economy.
  • Section 3: please, characterize your respondents with more statistics.
  • Subsection 3.4: all statistical approaches employed for the purposes of this study should be named there.
  • Subsection 5.2: you need to make some general recommendations to managers in a simple form.
  • The information is highly-complex and multiple. I strongly recommend to write a new section Conclusions bearing a list of 5-7 main findings (from Results and Discussion) formulated in relatively short phrases to be understood even by the non-devoted readers. Please, avoid abbreviations there. You also need to point out limitations and perspectives for further research there.

Good luck with revisions! I very hope to see your work published soon.

Author Response

Dear Reviewer,

Thank you for your comments. Appended to this letter is our point-by-point response to the comments. The comments are reproduced and our responses are given directly afterward in a different color (red).

This is example of very important and highly-interesting study. The paper itself is informative and stimulating, it is well-written and well-structured. The number of sources is enough. The figures and tables are proper. I think the paper needs only small amendments, after which it can be accepted.

  • Abstract: please, avoid abbreviations and tell a bit more about your findings.

Response: According to the reviewer’s suggestion, the abbreviations of the variables have been removed from the abstract and the following two paragraphs have been introduced on lines 29-34: The results of the study indicated that the variables Perceived Competitiveness and Perceived Risk have a significant impact on Intention to Use of Industry 4.0 processes while Perceived Vertical networking Solutions and Perceived Integrated Engineering Solution have a significant influence on the Intention to Use of Industry 4.0 solutions. In conclusion between Intention to Use of Industry 4.0 solutions and Benefits of Digital Transformation there is a positive and significant association.

  • Introduction: please, state that your study focuses on Romania and explain why.

Response: “In Romania” was introduced on line 59, “Romanian” was introduced on line 61 and the explanation for choosing Romania is integrated in lines 79-82 by introducing the phrase: ”We believe that this study will facilitate the transition of organizations to the implementation of Industry 4.0 and the benefits will provide managers with new directions for identifying priority criteria and real opportunities to make appropriate decisions in the business environment”.

  • Section 2: you need to add a subsection characterizing the Romanian economy.

Response: At lines 67-79, the following phrases have been introduced that characterize the evolutionary stage of Industry 4.0 in Romania: ”The stage of implementation of Industry 4.0 in Romania is incipient [10]. Currently, Romania has financing lines dedicated to the implementation of new innovative concepts and technologies such as Industry 4.0, 3DPrinting and Open Innovation which are supervised by the Ministry of Economy. Some companies operating in Romania, such as Deloitte, DHL, DB Schenker, Microsoft or Oracle, have made investments in: warehouse management systems (Warehouse Management System) or transportation management systems (Transportation Management System) but also applications that help track orders and transparency in supply chains. The largest companies have developed digital platforms [11], adopted IoT solutions [12] and augmented reality, developed 4G and 5G networks thus producing a revolution in IoT. There are significant development opportunities for Romania in terms of Industry 4.0 and the direction indicated becomes clear, which is why companies need to understand the importance and urgency of digitalization on which the success or failure of many of them depends”. On this occasion, three more reference notes were added.

[10] Türkeș, M.C.; Oncioiu, I.; Aslam, H.D.; Marin-Pantelescu, A.; Topor, D.I.; & Căpuşneanu, S., Drivers and Barriers in Using Industry 4.0: A Perspective of SMEs in Romania, Processes, 2019, 7(3), 153.

[11] Cokins, G.; Oncioiu, I.; Türkeș, M.C.; Topor, D.I.; Căpuşneanu, S.; Paștiu, C.A.; Deliu, D.; & Solovăstru, A.N. Intention to Use Accounting Platforms in Romania: A Quantitative Study on Sustainability and Social Influence. Sustainability, 2020, 12, 6127.

[12] Türkeș, M.C.; Căpuşneanu, S.; Topor, D. I.; Staraș, A.I.; Ștefan Hint, M.; & Stoenica, L. F., Motivations for the Use of IoT Solutions by Company Managers in the Digital Age: A Romanian Case. Applied Sciences, 2020, 10(19), 6905.

  • Section 3: please, characterize your respondents with more statistics.

Response: The following phrase was inserted on lines 360-362: ”The share of large enterprises in the surveyed sample is 15.27%. Most of them carry out activities in sectors such as industry 35.16%, trade (28.53%), services (22.19%) and others”.

  • Subsection 3.4: all statistical approaches employed for the purposes of this study should be named there.

Response: In line 365 the word "hypotheses" was replaced by the word "results", the word "tested" was replaced by the word "obtained", "ing" was deleted from the word "modeling" and "method" and added to line 366 ”Bootstraping and Blindfolding algorithm” and line 367 “to testing for group difference”.

  • Subsection 5.2: you need to make some general recommendations to managers in a simple form.

Response: The following sentence was introduced in the Conclusions section (lines 568-572): ”Given the research results, the main proposal for managing organizations is the use of Industry 4.0 processes and solutions to obtain benefits such as: streamlining workflow and improving productivity, increasing the speed of marketing innovative products, improving employee work tools for good communication and for performing work tasks and others”.

  • The information is highly-complex and multiple. I strongly recommend to write a new section Conclusions bearing a list of 5-7 main findings (from Results and Discussion) formulated in relatively short phrases to be understood even by the non-devoted readers. Please, avoid abbreviations there. You also need to point out limitations and perspectives for further research there.

Response: The following sentences were introduced on lines 513-519: ”The research results indicate that the variables perceived competitiveness (PC), perceived risk (PR) have a significant impact on Intention to Use of Industry 4.0 processes (IUP) while Perceived vertical networking solutions (VS) and Perceived integrated engineering solution (IS) have a significant influence on the Intention to Use of Industry 4.0 solutions (IUS). In conclusion we can say that between Intention to Use of Industry 4.0 solutions (IUS) and Benefits of digital transformation (BDT) there is a positive and significant association”.

Round 2

Reviewer 2 Report

Thank you for making the suggested corrections.